



# Evaluating uncertainty in sensor networks for urban air pollution insights

Daniel R. Peters[1], Olalekan A.M. Popoola[2], Roderic L. Jones[2], Nicholas A. Martin[3], Jim Mills[4], Elizabeth R. Fonseca[5], Amy Stidworthy[6], Ella Forsyth[6], David Carruthers[6], Megan Dupuy-Todd[1,7],
Felicia Douglas[5], Katie Moore[1,8], Rishabh U. Shah[1], Lauren E. Padilla[1], Ramón A. Alvarez[1]

[1]Environmental Defense Fund, 301 Congress Ave #1300, Austin, TX 78701, USA
[2]Yusuf Hamied Department of Chemistry, University of Cambridge, Cambridge, CB2 1EW, UK
[3]Air Quality and Aerosol Metrology Group, Atmospheric Environmental Science Department, National Physical Laboratory, Hampton Road, Teddington, Middlesex, TW11 0LW, UK
[4]ACOEM Air Monitors Ltd., Ground Floor Offices, C1 The Courtyard, Tewkesbury Business Park, Tewkesbury, Gloucestershire, GL20 8GD, UK
[5]Environmental Defense Fund Europe, 3rd Floor, 41 Eastcheap, London, EC3M 1DT, UK
[6]Cambridge Environmental Research Consultants Ltd., 3 King's Parade, Cambridge, CB2 1SJ, UK
[7]Now at: Clean Air Task Force, 114 State Street, 6th Floor, Boston, MA 02109, USA
[8]Now at: Clarity Movement Co., 808 Gilman Street, Berkeley, CA 94710, USA

*Correspondence to*: Daniel R. Peters (dpeters@edf.org)

**Abstract.** Ambient air pollution poses a major global public health risk. Lower-cost air quality sensors (LCS) are increasingly being explored as a tool to understand local air pollution problems and develop effective solutions. A barrier to LCS adoption is potentially larger measurement uncertainty compared to reference measurement technology. The technical
performance of various LCS has been tested in laboratory and field environments, and a growing literature on uses of LCS primarily focuses on proof-of-concept deployments. However, few studies have demonstrated the implications of LCS measurement uncertainties on a sensor network's ability to assess spatiotemporal patterns of local air pollution. Here, we present results from a 2-year deployment of 100 stationary electrochemical nitrogen dioxide ($NO_2$) LCS across Greater London as part of the Breathe London pilot project (BL). We evaluated sensor performance using collocations with reference
instruments, estimating ~35% average uncertainty (root-mean-square error) of the calibrated LCS, and identified infrequent, multi-week periods of poorer performance and high bias during summer months. We analyzed BL data to generate insights about London's air pollution, including long-term concentration trends, diurnal and day-of-week patterns, and profiles of elevated concentrations during regional pollution episodes. These findings were validated against measurements from an extensive reference network, demonstrating the BL network's ability to generate robust information about London's air
pollution. In cases where the BL network did not effectively capture features that the reference network measured, ongoing collocations of representative sensors often provided evidence of irregularities in sensor performance, demonstrating how, in the absence of an extensive reference network, project-long collocations could enable characterization and mitigation of network-wide sensor uncertainties. The conclusions are restricted to the specific sensors used for this study, but the results



give direction to LCS users by demonstrating the kinds of air pollution insights possible from LCS networks and provide a
blueprint for future LCS projects to manage and evaluate uncertainties when collecting, analyzing and interpreting data.

# 1 Introduction

Ambient (outdoor) air pollution is a leading contributor to human disease and mortality around the world, causing more than
four million premature deaths annually, with the greatest health burden in low- and middle-income countries (WHO, 2018;
HEI, 2020). Within cities, the burden of air pollution is not distributed equally, with significant spatial heterogeneity in
sources, concentrations, and exposures (e.g. Apte et al., 2017; Clark et al., 2014; Miller et al., 2020; Shah et al., 2020). Many
of the world's most populous and polluted regions are also those with limited air quality monitoring infrastructure, restricting
the potential for data-driven air quality management or public awareness campaigns (Pinder et al., 2019). Even in many
high-income countries, ambient air pollution monitoring is relatively sparse (e.g. Apte et al., 2017; US GAO, 2020).
Reference monitoring stations are state of the art in terms of accuracy and reliability and are required for regulatory reporting
(EU, 2008). However, they are costly ($\sim 10^4$ - $10^5$ USD).

Lower-cost air quality sensors (LCS) are increasingly being explored as an alternative or supplement to reference
monitors. LCS are orders of magnitude less expensive ($\sim 10^2$ - $10^4$ USD) and are therefore more suitable for dense
deployments. They are commercially available from numerous manufacturers, and the market is expanding rapidly. The
literature on LCS has primarily focused on technical evaluations of sensor performance in laboratory or field settings
(Castell et al., 2017; Duvall et al., 2016; Jiao et al., 2016; Karagulian et al., 2019; Kelly et al., 2017; Lewis et al., 2016; Mead
et al., 2013). Comprehensive reviews of sensor technology have identified common performance issues including drifting
baselines and cross-interference from other pollutants, as well as sensitivity to environmental conditions such as temperature
and relative humidity (WMO, 2021). The literature also presents a variety of approaches for improving the accuracy of
unprocessed sensor data including calibrations using collocations with reference instruments, in-field calibrations without
collocations, and machine learning techniques, among others (Kim et al., 2018; Munir et al., 2019; Sahu et al., 2021;
Spinelle et al., 2015; Zimmerman et al, 2018).

A growing literature on uses of LCS primarily focuses on scientific applications and proof-of-concept deployments.
Case studies have demonstrated the potential for LCS networks to provide data insights about a local air pollution
environment, including characterizing spatiotemporal trends in ambient air quality (Castell et al., 2018; Caubel et al., 2019;
Mead et al., 2013; Pope et al., 2018; Popoola et al., 2018) and improving air quality models through data fusion or
assimilation (Bi et al., 2020; Carruthers et al., 2019; Gupta et al., 2018; Lopez-Restrepo et al., 2021). While previous LCS
deployments often consider uncertainty of individual sensors relative to a reference instrument, we are unaware of network
deployments where the spatiotemporal observations have been directly compared to results from a reference network.

As LCS technology becomes more ubiquitous, there is growing interest from governments and civil society to use data
from LCS monitoring networks in air quality assessment and urban planning. To manage the inherent uncertainties of LCS,



guidance is needed on how users can evaluate sensor performance and decide on the most appropriate and robust uses of their data. In this work, we evaluate a sensor network's ability to characterize spatiotemporal air pollution patterns in the megacity of Greater London, by using data from an LCS monitoring network deployed as part of the Breathe London pilot project (BL).

London was an ideal study area for LCS evaluation due to the city's extensive network of reference air pollution monitors, as well as a range of additional tools including a detailed emissions inventory and high-resolution modelling, all of which contribute to an advanced understanding of historical and current air pollution (GLA, 2021). Further, while air pollution has improved in recent years, in 2019 an estimated 3600 to 4100 premature deaths were attributable to anthropogenic fine particulate matter ($PM_{2.5}$) and nitrogen dioxide ($NO_2$) in London alone (Dajnak et al., 2021) and pollutant

concentrations remain above UK and WHO guideline levels in many areas of the city (GLA, 2020a). In 2021, the Court of Justice of the European Union ruled that the UK has been exceeding legal limits of $NO_2$ since 2010 and that the government failed against its legal duties to put timely mitigation plans in place (The Guardian, 2021). This work focuses on $NO_2$ data, which was a key measurand of the project based on the local regulatory priorities.

We first evaluated the performance of a subset of $NO_2$ sensors that were collocated with reference instruments. The

uncertainties determined from these evaluations were then considered in the context of specific analysis applications, or "use cases", of LCS data including: long-term concentration trends, temporal concentration patterns (i.e. diurnal and day-of-week), and quantification of regional episodes of elevated air pollution. LCS network results were compared to results from an extensive network of London reference monitors, demonstrating the extent to which the BL network produced accurate spatiotemporal insights about air pollution, and illuminating how sensor uncertainties identified during collocations affected

the network's ability to characterize local air pollution.

While the BL LCS results show many areas of agreement with reference network data, with some areas of discrepancy, the comparisons are only representative of a selected sensor technology (electrochemical $NO_2$ sensors of a specific vintage from a specific supplier) deployed in a specific environment type; care should be taken in extrapolating results to other sensors and environments (i.e. differing pollution levels and weather conditions). Nevertheless, the methods and lessons

presented here can aid the design and operation of future LCS deployments by providing a blueprint for users to quantify and manage uncertainty in their own LCS data sets and explicitly consider the implications when investigating locally-relevant air pollution questions.

## 2 Methods

### 2.1 Monitoring devices

The BL $NO_2$ dataset includes data from 100 AQMesh units (Environmental Instruments Ltd., Firmware V 3.24), commercially available devices which have been previously tested and utilized by researchers and air quality managers (Fig. 1b) (AQMesh, 2021; AQ-SPEC, 2015; Castell et al., 2017). A detailed description of the AQMesh units can be found

elsewhere, e.g., Castell et al. (2017). AQMesh measurements of nitrogen dioxide ($NO_2$), the focus of this paper, relied on an Alphasense Ltd. $O_3$-filtered electrochemical sensor. The AQMesh devices in BL also measured nitric oxide (NO),

particulates ($PM_{2.5}$ and $PM_{10}$), carbon dioxide ($CO_2$), and 10 devices additionally measured ozone ($O_3$).

## 2.2 Network design and deployment

We deployed AQMesh units across Greater London (Fig. 1) in areas identified in consultation with the Greater London Authority (GLA), though final locations depended on obtaining permissions from site owners. We sought locations across a range of traffic levels and at varying distances from major roads and intersections, parks, residential areas, high-traffic

streets, and other commercial areas. In addition, we included monitoring at sensitive receptors including some primary schools and medical facilities.

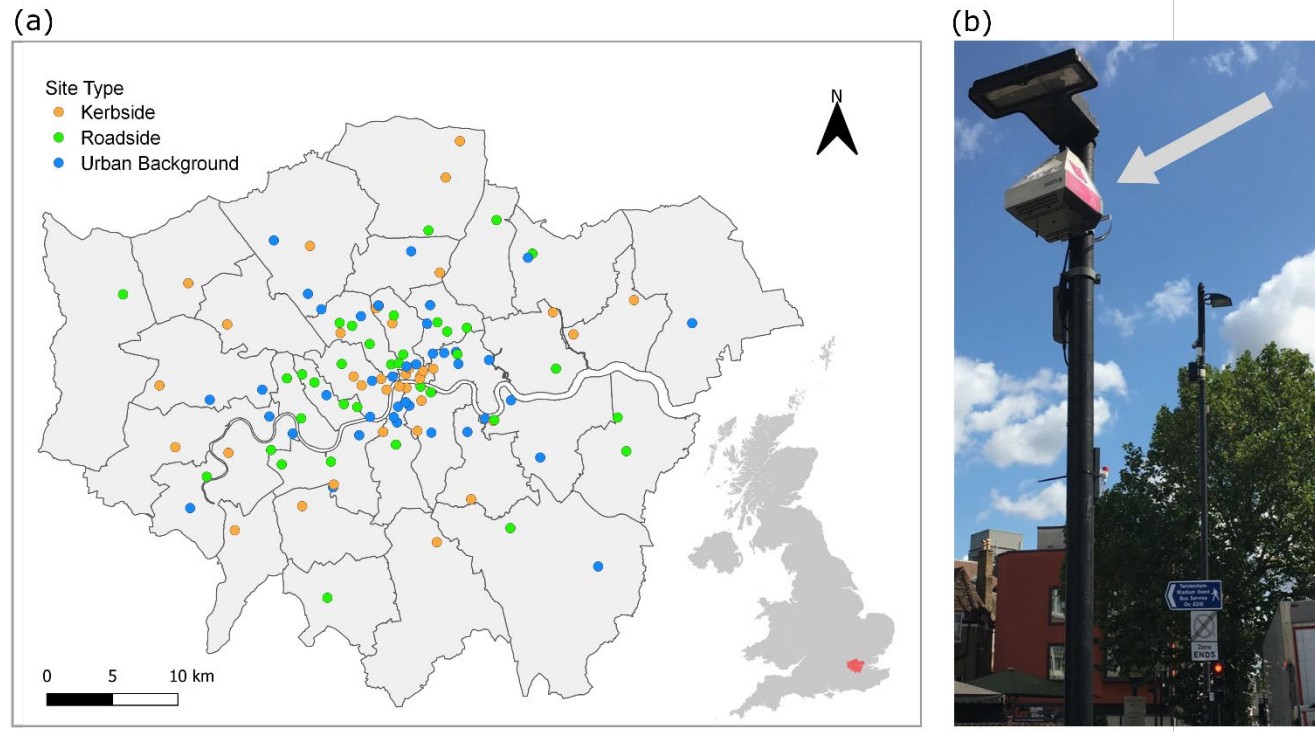

**Figure 1.** (a) BL network locations across Greater London. (b) Picture of BL AQMesh unit (indicated by arrow) installed at Kew Road, Richmond.

Each BL location was classified by site type (kerbside, roadside, or urban background) based on the local characteristics in accordance with GLA guidance for London air quality monitoring (GLA, 2018). Kerbside locations were usually within 1 meter (m) of a road and were expected to have high pollutant concentrations where traffic was the dominant source. Roadside locations were also situated near roads (usually <5 m) but were expected to be more representative of pedestrian exposure. Urban background locations were mostly sited within school yards away from dominant emissions sources such as

busy roads. BL AQMesh devices were often installed marginally higher (~3-4m) than London reference monitors (~2m) to



avoid physical tampering. Some monitors that were within one metre of the road were still classified as urban background or roadside based on judgement of local features including device height, positioning, and proximity to sources. While prevailing guidance recommends devices be placed away from structures, with 270-degree unobstructed flow, this goal was not achieved at many sites where the only option for installation and power supply was on a building façade (EU, 2008).

Thus, classifications are informative but somewhat imperfect.

Of the 112 AQMesh locations in the $NO_2$ dataset (number exceeds 100 because some sensors were relocated during the project), 36 sites were classified as kerbside, 36 as roadside, and 40 as urban background. The locations and site types are shown in Fig. 1a.

## 2.3 Data collection, processing, and QA/QC

We evaluated AQMesh measurements of $NO_2$ collected during the Breathe London pilot project from September 2018 through November 2020 (Breathe London, 2021a). The devices were set to take a measurement every 10 seconds and delivered averaged readings every minute (i.e., an average of six readings). These 1-minute data were transferred using a built-in GPRS modem to the manufacturer's (Environmental Instruments Ltd.) server in near real-time, where they were processed by the manufacturer using proprietary algorithms based on their factory testing, and are termed here as prescaled

data. Individual data points were accompanied by flags regarding sensor status. Data were then ingested into a data platform hosted by ACOEM Air Monitors Ltd., who also managed the monitor deployment, maintenance, and manual QA/QC process (Breathe London, 2020). Cambridge Environmental Research Consultants (CERC) applied a sensor-specific calibration gain and offset (see Sect. 2.3.1) to each device's 1-minute prescaled data to produce a calibrated dataset. CERC then filtered data for valid flags and high and low limits that screened out physically unrealistic concentration measurements

and averaged measurements to hourly time resolution using an 85% data capture threshold per hour. Manual inspection of sensor data was performed weekly to identify anomalous measurements. If a sensor malfunction was identified through QA/QC protocols, ACOEM Air Monitors Ltd. technicians intervened to mitigate the issue, usually through replacement of faulty sensors.

### 2.3.1 Sensor calibration

$NO_2$ sensors in the field (termed "candidate sensors" here) were calibrated using one of three methods: reference site collocation, transfer standard collocation, or remote network calibration method. For reference site collocations, a candidate AQMesh unit was installed alongside a reference monitor from the London Air Quality Network (LAQN) or UK Automatic Urban and Rural Network (AURN) (Fig. S1 shows a picture of an example reference site collocation). Transfer standard collocations relied on nine AQMesh devices that were periodically (every 2-4 months) collocated and calibrated against

reference monitors; these calibrated AQMesh units were then used as transfer standards and were collocated with so-called candidate AQMesh units in the field to determine the latter's calibration parameters. The duration of typical calibration collocations was 7-14 days (for both reference and transfer standard methods), though long-term collocations were also


conducted for further performance evaluation purposes. Calibration gain and offset parameters were obtained by performing a linear regression on the hourly averaged collocation timeseries after excluding statistical outliers. Calibration parameters were deemed valid and applied to the candidate sensor if the scaled collocation timeseries met statistical criteria of nRMSE < 50% (Eq. 3) and $R^2$ > 0.7 (Eq. 4), which ensured that sensor performance was sufficient to calculate robust calibration parameters and effectively excluded periods where the $NO_2$ variability was too low to provide a meaningful test of sensor gain and offset.

The remote network calibration method is a novel approach, developed and applied by the University of Cambridge project team, that remotely derives unit-by-unit calibration parameters for the entire sensor network in lieu of physical collocations. The algorithm uses a spatial scale separation methodology described in previous work (Heimann et al., 2015; Popoola et al., 2018) to calibrate sensors in relation to each other when pollution levels are consistent across the network and obtains traceability (connection to reference standard with known uncertainties) from a single calibrated reference monitor (Popoola et al., in preparation). For BL, a single (site dependent) calibration was performed using the period May-Dec 2019 and applied to the entire dataset. When multiple valid calibration options were available for a specific AQMesh sensor in the network, a decision tree was used which prioritized: i) reference site collocation (n=11) ii) transfer standard collocation (n=73) and iii) network calibration (n=38); the total number of calibrations applied exceeded the number of devices because failed sensors were replaced and re-calibrated.

**2.3.2 Ozone cross-interference correction**

A long-term upward drift in BL $NO_2$ sensor measurements was identified and assessed to be most likely caused by an ozone cross-interference. A correction was applied to the hourly $NO_2$ dataset that subtracted a fraction of the derived site-specific $O_3$ concentration from the scaled $NO_2$ readings. Site-specific $O_3$ was deduced using upwind background reference $O_3$ measurements; under low-NOx conditions (<10 ppb) the site-specific $O_3$ was assumed to be the upwind background $O_3$ concentration, otherwise it was assumed to be the difference between background $O_3$ and the site-specific NO concentration. Because the effect appeared to increase as a sensor aged, the cross-sensitivity correction for ozone was assumed to start at 0% upon initial sensor deployment and exponentially increase to a maximum of +18% of estimated site-specific ozone concentrations 6 months later. Figure S2 illustrates the effect of the correction on BL network mean $NO_2$ concentrations throughout the campaign. Except for the short-term collocation analysis results (Fig. 2), the results presented throughout this paper use the scaled hourly-average ozone-corrected dataset.

Detailed documentation of the static network QA/QC procedures are available in the project QA/QC manual in the Breathe London Technical Report (Breathe London, 2020).

**2.4 Reference and meteorological data**

Hourly $NO_2$ and $O_3$ concentration data were downloaded for 105 reference monitors within Greater London that were classified as kerbside (n=12), roadside (n=62), or urban background (n=31) using the R openair package (Carslaw and



Ropkins, 2012). These monitors, which we refer to collectively as the "reference" network, include sites from multiple overlapping UK networks including the London Air Quality Network (LAQN), Air Quality England network (AQE), and Automatic Urban and Rural Network (AURN). At the time of download (9 June 2021) reference data were fully ratified for 69 sites. At 36 sites, some 2020 data were categorized as provisional and are thus subject to change during the ratification process. Hourly ambient air temperature observations at London Heathrow Airport, located within the Greater London study

region and ~25 km west of Central London, were accessed from the National Oceanic and Atmospheric Administration (NOAA) Integrated Surface Database (ISD) via the R worldmet package (NOAA, 2021; Carslaw, 2020).

**2.5 Sensor performance statistics**

The reference site collocations described in Sect. 2.3.1 were also used to evaluate sensor performance. A total of 98 collocations were performed between a LC sensor and a reference monitor, including 10 sensors that were collocated more

than once and 2 sensors that were collocated for long-term periods of >80 weeks. The statistics in Eq. (1-4) were used to evaluate sensor performance during reference site collocations (a representative example of collocation results is shown in Fig. S3). The following statistics were calculated from hourly timeseries data for each individual collocation of $n$ hours duration:

Mean Bias Error (MBE) = $\frac{1}{n}\sum_{i=1}^{n}(Sen_i - Ref_i)$,                                                                                   (1)

Root Mean Square Error (RMSE) = $\sqrt{\frac{1}{n}\sum_{i=1}^{n}(Sen_i - Ref_i)^2}$,                                                          (2)

Normalized Root Mean Square Error (nRMSE) = $\frac{\sqrt{\frac{1}{n}\sum_{i=1}^{n}(Sen_i - Ref_i)^2}}{\overline{Ref}}$,                    (3)

Coefficient of Determination (R$^2$) = $1 - \frac{\sum_{i=1}^{n}(Sen_i - Ref_i)^2}{\sum_{i=1}^{n}(Sen_i - \overline{Sen})^2}$,            (4)

where *Sen* represents the BL sensor measurement and *Ref* represents the observed reference measurement.

**2.6 ADMS-Urban modelling data**

The ADMS-Urban air pollution dispersion model was used to simulate 2019 hourly NO$_2$ concentrations at BL and reference network monitoring locations (McHugh et al., 1997). The model used traffic flows and speeds and 1km gridded emissions of NO$_2$ from the London Atmospheric Emissions Inventory (LAEI) 2013 dataset (published in 2016), interpolated to 2019 from the 2013 base year and 2020 future predictions, combined with road traffic emissions factors from the Emission Factor Toolkit (EFT) v8 for 2019 and real-world adjustment factors to calculate road source emissions. The model includes

atmospheric chemistry as well as complex urban effects including street canyons and urban canopy. Individual monitoring sites were modelled as discrete receptors with the appropriate position and height. NO$_2$ sources from outside the modeled domain were represented using hourly background concentrations at one of four rural AURN (Automatic Rural and Urban Network) stations located outside Greater London, based on which station was upwind at that hour, and hourly



meteorological data was used from London Heathrow Airport. The modelling scenario ("Hotspot 2019") includes weekday
diurnal emissions patterns to represent variations in traffic flow and improvements to LAEI traffic flow. Additional details
on the ADMS-Urban model and Hotspot 2019 scenario are available in the Breathe London Technical Report (CERC, 2021).
To calculate the modelled difference between BL and reference network means for the year 2019 (Sect. 3.2.1), we selected
all monitor-hours with valid mod-obs pairs (i.e. a valid modelled and observed concentration existed at that hour) for all
reference and BL sites analyzed in this manuscript. The modelled 2019 means were calculated for each network from the
pooled monitor-hours.

## 3 Results and Discussion

### 3.1 Network performance

### 3.1.1 Data capture

The BL network generated nearly 1.5 million hourly calibrated $NO_2$ measurements from 100 devices at 112 locations over
the course of the 26-month pilot campaign. The number of sensor locations producing valid, calibrated data gradually
increased over the first seven months (Fig. S4). The initial delay in network data capture was caused by logistical challenges
faced at the outset of the project including obtaining permissions for monitor deployment and conducting calibrations for
each sensor. By the spring of 2019 the majority of the network was operational, and generated valid data for the remainder of
the project, though the total number of $NO_2$ sensors pods producing valid data fluctuated due to redaction of flagged data and
the downtime of sensors that failed during the project before replacement and re-calibration were performed. In total, 35 $NO_2$
sensors were replaced due to failure, with most failures occurring during the winter. Additional considerations and lessons
learned for stationary sensor network setup and maintenance are discussed in the Breathe London Blueprint (Breathe
London, 2021b).

### 3.1.2 Measurement uncertainty of calibrated sensors

Figures 2 and 3 present measurement uncertainty statistics for calibrated BL LCS based on short-term (typically 7-14 days)
and long-term (>80 weeks) reference collocations. Both analyses quantify uncertainty of sensor measurements that were
calibrated based on results of a prior reference site collocation (Sect. 2.3.1). These results allow us to evaluate the
effectiveness of the project's QA/QC procedures (including calibration) since each repeat collocation serves as an
independent test of the project-long uncertainties of sensors that were calibrated during a discrete time period.

Figure 2 shows evaluation results for 10 calibrated sensors that were collocated for subsequent short-term periods
(typically 7-14 days) that began 1-84 weeks after an initial reference site calibration period. These subsequent collocation
periods (n=35) were used to estimate calibrated sensor uncertainty compared to reference measurements (e.g., unit 99 was



calibrated based on the first reference site collocation in October 2018; uncertainty statistics in Fig. 2 were calculated from the second and third reference collocations which occurred in April and July of 2019).

A median $R^2$ of 0.79 indicates that calibrated sensors effectively captured changes in NO₂ concentrations that were measured by reference instruments. The median MBE was 8.0 µg m⁻³ (23% of mean concentration) with a range of -19 to 34 µg m⁻³ (-37 to 121% of mean concentration); and the median normalized RMSE was 35% (range of 16 to 189%). The large range of biases exhibited by individual sensors and the systematically high median bias of the collocated sensors reveal variability in the consistency of sensor response over time (and under different meteorological conditions), and serve to

assess the robustness of initial sensor calibrations when applied to a longer timeseries. However, we note that uncertainty statistics in Fig. 2 are calculated from sensor data that was not corrected using the ozone cross-interference correction (Sect. 2.3.2) and thus represent an upper bound of the BL network uncertainty.

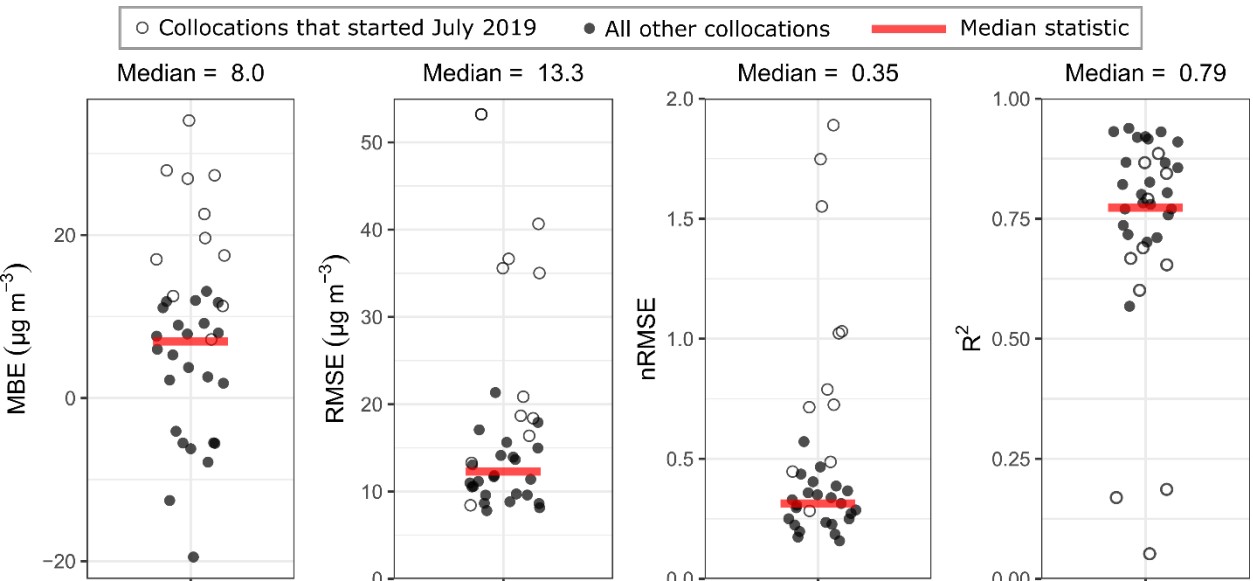

**Figure 2:** Performance of calibrated sensors during short-term (typically 7-14 days) collocations with reference instruments. Unfilled circles are collocations that started in July 2019 during periods of elevated temperatures. Statistics calculated from hourly measurements (eq. 1-4).

The Fig. 2 results and summary statistics are affected by a group of outlier collocations (unfilled circles in Fig. 2) that started July 2019, during which most sensors exhibited higher measurement error and poorer correlation to reference

measurements. The 8 collocations with the highest normalized RMSE (>70%) all occurred during July 2019 (Fig. S5). Additionally, 7 of these July 2019 collocations had $R^2$ values below 0.7, meaning they would have failed the statistical screening criteria used for determining valid collocation calibrations (Sect. 2.3.1). During this month, we observed high-biased sensor measurements when local air temperatures were above 20-25°C which we discuss further below. With the July 2019 collocations (n=11) excluded, the median nRMSE and MBE of the remaining collocations (n=24) improve to 30% and

4.5 µg m⁻³ respectively, and the median $R^2$ increases slightly to 0.81.





**Figure 3:** Performance of two calibrated sensors during long-term reference collocations. Sensors were calibrated using linear regression against the reference instrument during a two-week collocation directly preceding the evaluation period (calibration period not shown). (a) Daily mean NO₂ concentration timeseries comparison of BL sensor and reference monitor measurements. (b) Monthly MBE (eq. 1) and RMSE (eq. 2) statistics of hourly BL sensor measurements compared to reference measurements. (c) Scatter plot and statistics (eq. 1-4) comparing hourly BL sensor (x-axis) and reference monitor (y-axis) measurements for entire evaluation period.





Figure 3 presents the collocation timeseries and monthly error statistics between calibrated BL sensor and reference monitor measurements during two long-term (>18 month) collocations, where the sensor measurements are calibrated based on the collocation results during the two-week period directly preceding the extended evaluation period. While the aggregate MBE of both collocations is small (<2 µg m$^{-3}$), BL sensors exhibit biases that vary seasonally relative to reference measurements; monthly MBE of sensors ranges from -2 µg m$^{-3}$ to +8 µg m$^{-3}$ for unit 17 (-7% to +31% of the monthly mean concentration) and from -6 to +14 µg m$^{-3}$ for unit 83 (-16% to +72% of the monthly mean concentration). The drifting sensor response follows the same seasonal pattern for both long-term collocations, with the highest bias occurring during summer months and peaking during August 2020. Variations in monthly RMSE error are largely driven by sensor bias; nRMSE is highest during summer months, corresponding to peak BL sensor bias. Fig. S6 further illustrates the occurrence of high-biased BL sensor measurements during hours when the local air temperature exceeded 20-25°C. Aside from the seasonal variation in sensor bias and error, the initial calibrations seem to hold over the duration of the 18-month collocations.

While the results presented above quantify uncertainty of sensors calibrated using reference collocations, the data use cases in the following sections also include sensor data calibrated using two additional approaches when sensors could not be collocated at reference sites, as described in the methods (Sect. 2.3.1): transfer standard calibration and network calibration method. The transfer standard method is more difficult to validate because collocations occur at BL sites in the field instead of at reference sites. The uncertainty of this method is expected to be marginally higher than the direct reference site collocations in Fig. 2 due to the additional step where the calibration is transferred between BL AQMesh units. A high level of precision and consistency in response across BL NO$_2$ sensors (R$^2$ = 0.94; nRMSE = 0.1; Fig. S7) gives confidence that calibrations would transfer effectively between units. An evaluation of the performance of the independent network calibration method is included in Popoola et al. (in preparation). In brief, the estimated uncertainty of sensor measurements scaled with network method is broadly similar to the uncertainty of reference collocation-calibrated sensors (~30% median nRMSE). The results in Figs. 2 and 3 suggest that regardless of the method used for calibration, measurement uncertainty of sensors calibrated during a discrete period will be largely driven by the variability in the sensor performance over time. Enhanced QA/QC such as application of the remote network calibration method on a near-continuous basis or seasonal bias corrections such as shown in Fig. S9 (see Sect. 3.2.1) could minimize variations in measurement uncertainty due to sensor performance.

For a long-term measurement campaign using sensors, evaluation against reference measurements should be performed throughout the course of the project. The evaluation results above point to the ability of BL sensors to accurately reproduce changes in NO$_2$ concentrations captured by the reference monitors (high R$^2$ values) with average uncertainty (nRMSE) of ~35%. However, our results also show that seasonal biases due to time-varying effects of environmental interferences can lead to larger uncertainties (>100% nRMSE) during periods when local air temperatures reached above 20-25°C. This characterization of sensor uncertainties can inform how results from the BL LC sensor network are interpreted, ensuring derived insights are robust (e.g., differentiating between high-biased sensor artifacts and elevated NO$_2$ concentrations).



We next present a series of analytical use cases to evaluate the applicability of BL NO₂ LCS network results for deriving insights about the local air pollution environment. Results from each use case using BL data are compared against results generated from reference network data. In addition, the collocation sensor evaluations presented above are used to assess BL network uncertainty and interpret differences between BL and reference network results.

**3.2 Use case validation**

**3.2.1 Regional pollution load and time trends**

We first examine the ability of the BL sensor network to characterize trends in the regional (Greater London) pollution load, by comparing monthly mean NO₂ concentrations of the BL network with the reference network results (Fig. 4). We note two major events during the measurement campaign which are expected to impact NO₂ concentrations: i) introduction of the Ultra Low Emission Zone (ULEZ), which became effective on 8 April 2019, imposed tolls to discourage entry of older, higher emitting vehicles into Central London, with increasing fractions of compliant vehicles and fewer vehicles overall observed in the zone through calendar year 2019 (GLA, 2020b), and ii) COVID-19 pandemic restrictions beginning March 2020, including social distancing measures and stay-at-home orders, disrupted activity patterns throughout Greater London.

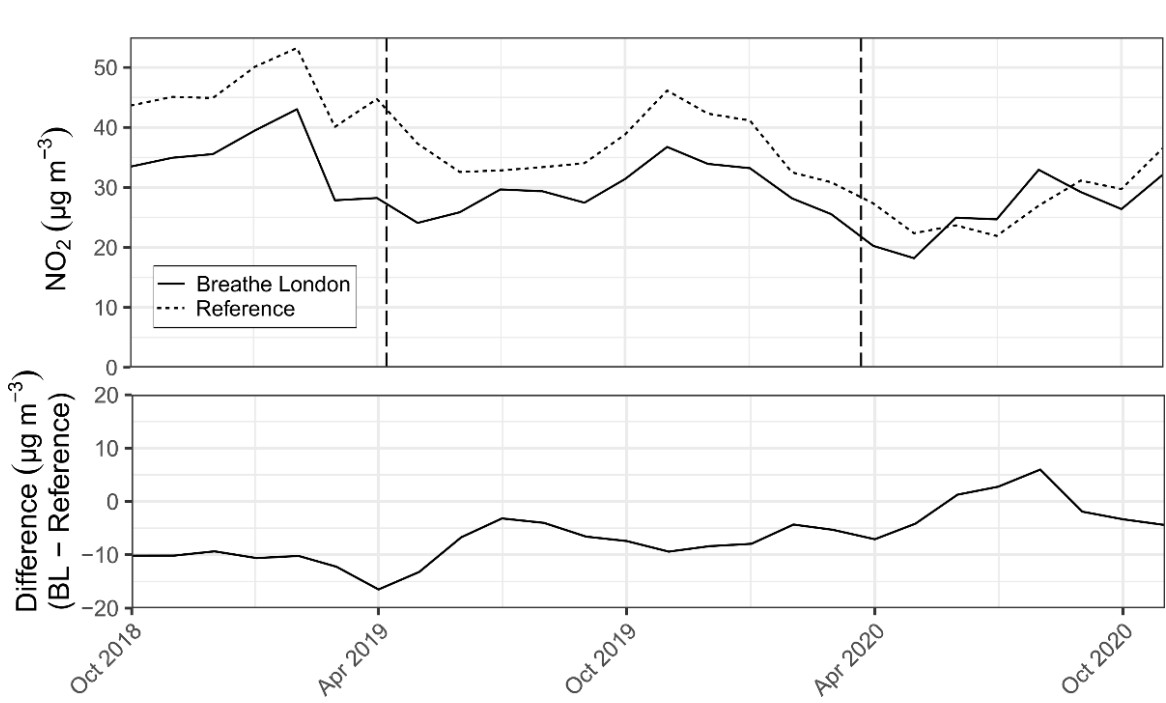

**Figure 4:** Comparison of monthly mean NO₂ concentrations for the BL (n=100) and London reference (n=105) networks. Bottom panel shows difference between networks. Vertical lines denote Ultra Low Emission Zone (ULEZ) start date (8 April 2019) and the start of the first Covid-19 lockdowns (23 March 2020).



The BL network tracked the reference network trend while exhibiting lower mean concentrations for most of the campaign (on average 7 µg m$^{-3}$ lower throughout campaign; 9 µg m$^{-3}$ during 2019). We attribute this partially to differences in location, site types, and sampling points (height, distance to road, road traffic volume, e.g.) between the networks and this

is confirmed through comparisons of measured and modelled concentrations using the ADMS-Urban air pollution dispersion model (described in Sect. 2.6). Modelled network mean NO$_2$ concentrations for 2019 at reference network monitoring site receptors were 5 µg m$^{-3}$ higher (~15%) than the modelled mean concentrations at BL receptor locations. Because the model only predicts 55% of the difference between the two networks, we examined the model-network comparisons more closely. The model exhibits little systematic bias at reference sites (<1 µg m$^{-3}$; see Fig. S8). By contrast, the mean of modelled

concentrations was higher than that observed at the BL sites by 6 µg m$^{-3}$, with the difference driven by BL sites with the lowest observed concentrations (Fig. S8). We note that 16 BL sites exhibited lower concentrations than the 20 µg m$^{-3}$ minimum observed by the reference network, so we cannot rule out the possibility of low sensor bias in a portion of the BL network. In sum, we are unable to fully resolve the cause of the systematic difference between modelled and observed BL concentrations, although it may have contributions from uncertainty in sensor network measurements (and underlying

QA/QC) and model uncertainty.

Both networks show a downward year-on-year trend in NO$_2$ concentrations and seasonal variability with peak concentrations in the winter. However, BL NO$_2$ means exhibit local maxima in July and August when reference network measurements are lowest. This effect is the most pronounced in summer 2020 which is the only time when the BL network average exceeds that of the reference network. This bias of the BL network compared to reference network trends during

summer months is likely due in part to a systematic high bias in the BL network's NO$_2$ sensors coinciding with local air temperatures above 20-25°C, an effect which was evident during collocations with reference monitors (Fig. 3, Figs. S5 and S6). However, spatially varying NO$_2$ pollution trends (e.g., Covid-19 restrictions having a larger impact on emissions at specific monitoring sites or city neighbourhoods) may have also affected the two networks differently and contributed to the converging network means towards the end of the BL campaign.

The long-term collocations (Sect. 3.1.2) were used to quantify seasonal changes in sensor bias and could serve as a basis for an empirical correction to the Fig. 4 BL network timeseries to improve the accuracy of the LCS results. This correction relies on the performance results being consistent across the network; the high precision between AQMesh units in our transfer standard locations (median R$^2$ = 0.94; Fig. S7) supports this assumption for the BL project. In Fig. S9 we show the Fig. 4 BL timeseries with a monthly bias correction based on the long-term collocations that would largely mitigate the

seasonal irregularities in the BL timeseries compared to the reference network.

### 3.2.2 Temporal pollution patterns

We next compare the recurring temporal patterns in NO$_2$ concentrations measured by the BL and reference networks (Fig. 5).



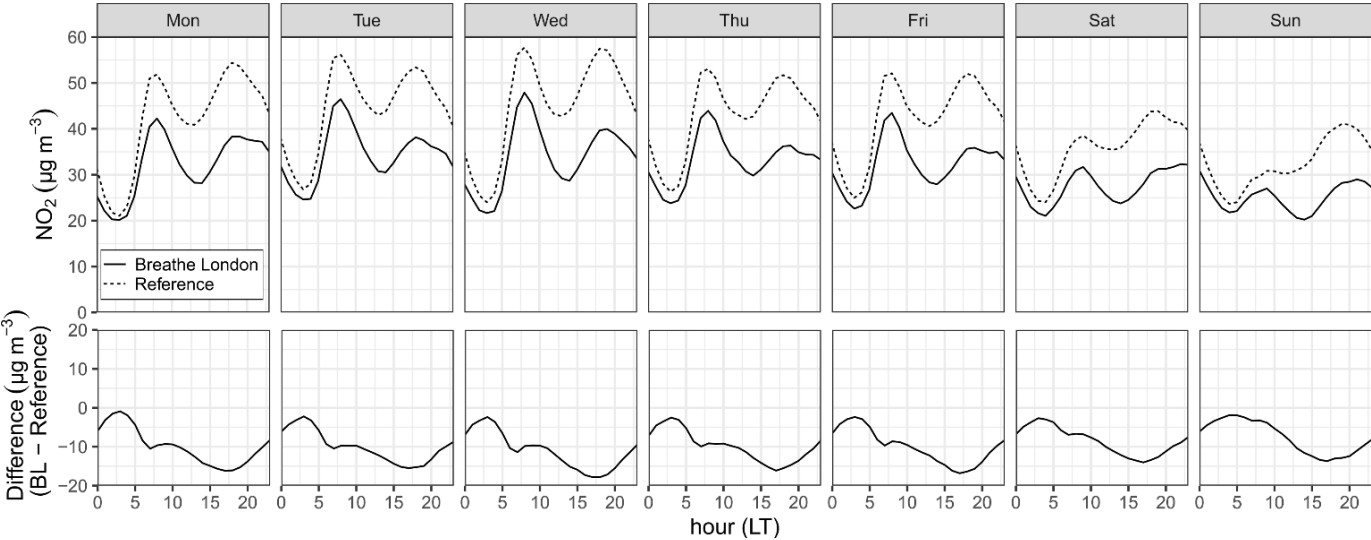


**Figure 5:** Network mean diurnal and day-of-week NO₂ concentration patterns in Greater London, as measured by the BL (n=99) and reference (n=105) networks during the pre-Covid-19 period of the BL project (1 Oct 2018 through 29 Feb 2020). Bottom panel shows difference between networks.

The BL network captures diurnal and day-of-week patterns with three key differences from the reference network. First, BL network mean concentrations are ~10 µg m⁻³ (23%) lower than the reference network result. Most of this difference was predicted in the modelling exercise discussed in Sect. 3.2.1, with additional contribution from uncertainty in sensor measurements. Second, BL network mean concentrations show a reduced diurnal range compared to the regulatory network (i.e. though daytime average BL concentrations are lower, night-time values are similar to the reference network). This

behavior may be due to previously discussed differences in site characteristics (higher sensor placement and lower traffic volume at near-road sites, e.g.) yielding reduced heterogeneity in site types across the BL network, which as a whole appears to be measuring diurnal pollution patterns that are more in line with urban background reference sites (Fig. S6). A similar effect is observed in a comparison of near-road (kerbside & roadside) and urban background reference sites, where the concentration difference was smallest during late night/early-morning hours (Fig. S10). A third key difference in diurnal

day-of-week concentration patterns is the magnitude of the evening peak, which is consistently lower than the morning peak in the BL network. On Wednesdays, for example, the reference network evening peak reached 57 µg m⁻³ at 6 PM LT while the BL network reached 40 µg m⁻³ at the same time; other weekdays similarly have the largest difference in network mean concentrations during the evening rush hour peak. We have not identified a mechanism to explain this difference, which is evident, to a varying degree, throughout the year (Fig. S11).

The BL network was able to accurately characterize timing of peaks and troughs in diurnal variability as well as capture differences in weekday and weekend pollution levels. Uncertainties in the precise magnitudes of some features remain, with the evening peak registering relatively lower in the BL network.

### 3.2.3 Site type differences in diurnal pollution patterns

We next examine the ability of the BL network to detect differences in diurnal $NO_2$ concentration patterns at different
monitoring site types. Figure 6 shows the weekday diurnal averages for the BL and reference network at near-road (kerbside
& roadside) sites compared to urban background sites.

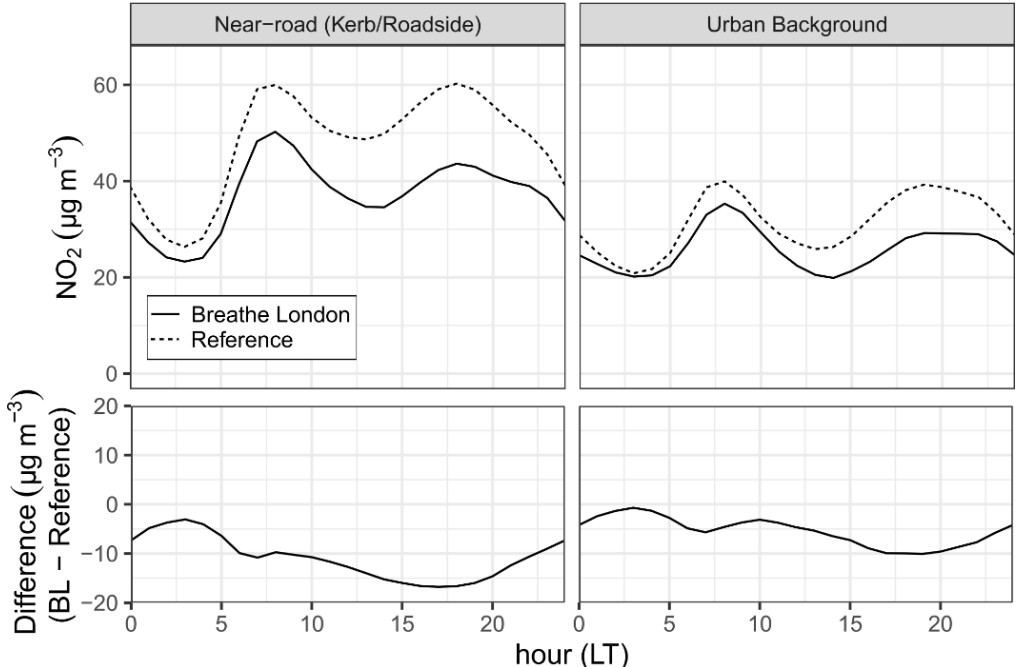

**Figure 6:** Weekday diurnal mean $NO_2$ concentrations in Greater London as measured by the BL (n=70 near-road; n=40 urban background
– number of locations exceeds 100 because some devices were placed at multiple locations during the campaign) and reference (n=72
near-road; n=31 urban background) networks during the pre-Covid-19 period of the BL project (1 Oct 2018 through 29 Feb 2020), at two
different site classification groups: near-road (left; includes sites classified as kerbside and roadside) and urban background (right). Bottom
panel shows difference between networks.

In the morning, near-road concentrations peaked at 8-9 AM LT in both the BL and reference networks, reaching 60 µg
$m^{-3}$ in the reference network and 50 µg $m^{-3}$ in the BL network. The time of the evening peak was also consistent between
networks, occurring at 6-7 PM LT and reaching 60 µg $m^{-3}$ in the reference network compared to a lower peak of 44 µg $m^{-3}$ in
the BL network at the same time. In both networks, the evening peak in concentrations occurred one hour later (7-8 PM LT)
at background sites than near-road sites. The greatest difference between BL and reference means at both near-road and
urban background sites occurred during the evening peak in $NO_2$ concentrations; this feature was identified in the network-
wide trends in the prior section.

At this aggregate level, the lower-cost network captures similar diurnal features and effectively differentiates between
pollution levels and time-of-day trends at urban background and near-road sites.



### 3.2.4 Hotspots and spatial heterogeneity

Here we discuss the application of BL LCS data for identifying hotspots and characterizing spatial heterogeneity in $NO_2$ concentrations, using a case study where BL sensor measurements led to identification of an air pollution hotspot. During the

first winter of the project (Dec 2018 through Feb 2019), a BL sensor deployed at Holloway Bus Garage measured mean weekday $NO_2$ concentrations of 77 µg m$^{-3}$, 89% higher than the BL network weekday mean of 41 µg m$^{-3}$ (Fig. S12). Though the concentration gradient (between Holloway Bus Garage and the BL network mean) was larger than the typical sensor uncertainties (~35% nRMSE), and occurred during winter months when large positive biases were not observed during collocation evaluations, additional steps were taken to establish confidence that the local pollution levels were accurately

characterized and not sensor artifacts. Two additional BL sensors were deployed in the area and a follow-up transfer standard collocation was performed which verified the accuracy of the deployed pod's calibration factors.

The BL monitoring at Holloway Bus Garage ultimately led to corrective action by local authorities, and this successful example demonstrates the potential value of LCS for identifying air pollution hotspots. The case study also emphasizes the need for rigorous verification of measurements from an individual sensor. The collocation analyses quantified a wide range

in the bias of BL sensors over the course of the project, as well as uncertainty in the consistency of sensor performance over time (Figs. 2 and 3). Therefore, especially for concentration gradients of similar magnitude to the estimated uncertainty of the sensors, there is a need for caution when analyzing site-specific data; we established confidence in the LC sensor hotspot characterization through the deployment of additional LC sensors to verify results.

### 3.2.5 High pollution episodes

Here we test the viability of the BL network to detect short- to medium-term (hours to days) episodes of elevated $NO_2$ concentrations using a well-characterized historical air pollution event in December 2019 (LAQN, 2019). Weather conditions in Greater London resulted in the formation of a strong temperature inversion that caused a build-up of primary pollutants including $NO_2$ in the layer of colder air close to the ground, with pollution peaking at morning rush hour on December 4$^{th}$ (LAQN, 2019). Fig. 7 compares the hourly mean $NO_2$ concentrations as measured by the BL and reference

networks for the week of the pollution episode.

The BL network detected a short-term regional build-up of pollution with a temporal profile that provides excellent comparability with the reference network result ($R^2$ = 0.96) and corresponds to the London Air Quality Network's published report about the event. The highest peak occurred during late rush hour (9-10 AM) on the morning of December 4$^{th}$ with the BL network registering a peak of 87 µg m$^{-3}$ compared to 103 µg m$^{-3}$ for the reference network during the same time period.

Another smaller peak occurred when evening emissions were trapped on the 4$^{th}$, and the event subsided when the inversion broke near midnight on the 4$^{th}$. The BL lower-cost network captures the basic features of the event although there is a low bias compared to the reference sites (nRMSE = 23%; MBE = -11 µg m$^{-3}$), partially explained by the different site types (see





Sect. 3.2.1). Additionally, the BL network could provide such information in near-real-time, making it a viable tool for rapid dissemination of air quality alerts.


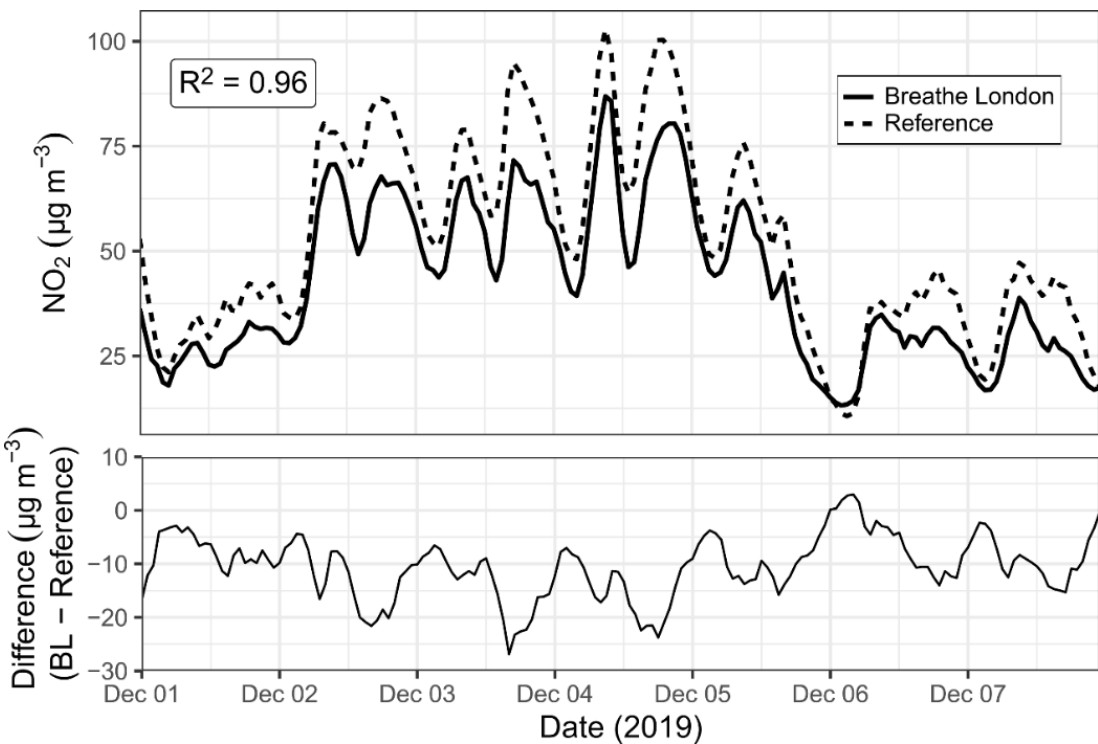

**Figure 7:** Hourly network mean $NO_2$ concentrations for the BL (n=85) and reference (n=105) networks during a high-pollution episode in December 2019. Bottom panel shows difference between networks.

However, we note that the network was less effective in characterizing pollution events during periods of poorer sensor performance. In Fig. S13 we present a more cautionary case study during July 2019. We demonstrated in Sect. 3.1.2 that collocated BL sensors produced high-biased measurements during periods when local air temperatures reached above 20-25°C with worst-case nRMSE exceeding 100% (dominated by positive bias). Figure S13 shows an instance where this effect leads to an overestimation of regional $NO_2$ pollution levels using the BL network (e.g. the BL network mean during daytime

hours on 25 July 2019 is 91 µg m$^{-3}$ compared to the reference network mean of 65 µg m$^{-3}$, a 40% positive bias across the network). Due to the extensive reference network in London and frequent short-term as well as ongoing long-term BL sensor collocations, we were able to identify the apparently anomalous BL sensor behavior under these environmental conditions which resulted in a systematic positive bias across the network. However, in cases with limited reference monitoring infrastructure, the BL measurements could have led to an overestimation of the magnitude of the pollution event in question.

While technological and methodological solutions to address this sensor issue are viable, another project with different technologies or environmental characteristics may experience different effects, illustrating the importance of rigorous data validation and uncertainty evaluation in the context of each new application of LCS technology.



## 3 Conclusions

In a LCS deployment, careful evaluation of sensor performance (which may vary between projects due to, e.g., specific
sensor technology, firmware, local meteorology and pollution characteristics, among others) maximizes the value of the data
by informing how it should be processed, analyzed, and interpreted. Robust uncertainty characterization and validation
against reference instruments equips the user to take full advantage of data including: i) developing corrections (see Fig. S9
presenting the BL network timeseries with a potential correction derived from collocation evaluation results), ii) excluding
measurements during conditions where sensor performance might be compromised, or iii) ensuring analyses are appropriate
based on the data quality. By contrast, we have shown that without a detailed understanding of variations in sensor
performance across a campaign (see Fig. 3b illustrating temperature related drift), biased sensor measurements at some
moments during the project could have led users to overestimate pollution levels, or over longer timeframes, miss trends in
concentration patterns. Our findings emphasize the importance of monitoring sensor performance for the duration of a
measurement campaign, as even pre- and post-campaign sensor evaluations may not have detected the seasonal changes in
sensor performance that our repeat (Fig. 2) and long-term (Fig. 3) collocations allowed us to quantify. A cost-effective
calibration approach such as the remote network calibration method can also be valuable for tracking and improving sensor
performance over time by providing periodic calibrations and assessments of network performance, although a single point
calibration was used here.

Our results also demonstrate how LC sensors could be used in a city with more limited existing monitoring
infrastructure than in London. The BL network generated a series of insights about air pollution in London that we compared
to reference network results, and we found that the BL LCS network characterized many $NO_2$ trends and patterns effectively
including year-over-year concentration trends, timing of diurnal peaks, weekday-weekend concentration gradients, and
profiles of short- to medium-term periods of elevated pollution. We also showed how BL sensor uncertainties, which were
evaluated using collocations at three London reference monitors, limited the LCS network's ability to capture precisely some
features of air pollution trends – emphasizing that especially in a place without an extensive reference network, it is
advisable to have at least one reliable reference instrument as a basis for ongoing LC sensor calibration and uncertainty
evaluation. We also note that the use of representative reference collocations (i.e. keeping one or two units at reference sites
throughout the project) to estimate network performance relies on the testable assumption that sensors are highly precise
across the network.

The sensor uncertainties and data use cases that we have evaluated are specific to the sensor technology and firmware
used as well as the local environmental characteristics in London. In London, environmental effects significantly impacted
data quality including frequent winter-time sensor failures and high measurement artifacts occurring when local ambient air
temperatures exceeded 20-25°C, indicating that sensor performance could vary in other cities with different source patterns
and meteorology. Additionally, the current absence of a performance standard for LC sensors exposes the end user to risks in
the sensor selection process, making it advisable for each implementation of LCS technology to perform its own



performance evaluation. Our approach can provide a roadmap for future LCS deployments to maximize data quality and confidence in resulting insights by following robust QA/QC protocols, most notably the tracking of representative sensor performance for the duration of the project via direct traceability to reference measurements.

**Data availability**

BL network data are available on the OpenAQ platform: https://openaq.org. London reference monitor data can be accessed using the R openair package (Carslaw and Ropkins, 2012). Meteorological data are available from the NOAA Integrated Surface Database (NOAA, 2021). Collocation data are available upon request from the corresponding author.

**Author contributions**

EFonseca, MD, FD, and KM managed the project including coordinating sensor deployment and collocations. JM oversaw installation, operation, and maintainance of the sensor network. NM hosted sensor system deployment. RJ and OP developed and applied the remote network calibration method and ozone correction to the hourly $NO_2$ dataset and were involved in data curation. AS, EForsyth, and DC applied the QA/QC procedure to the raw 1-minute AQMesh measurements to produce a
calibrated hourly dataset, curated data sets and metadata, and developed and ran model simulations. RA, RJ, DC, and NM supervised research. RA, DP, and LP formulated research goals for this manuscript. DP prepared the manuscript including formal analysis, visualization, and writing. All co-authors contributed to reviewing and editing.

**Competing interests**

During parts of the Breathe London pilot project, KM and JM were employed at commercial sensor providers (Clarity Movement Co. and ACOEM Air Monitors Ltd., respectively) and the University of Cambridge had a commercial arrangement with AQMesh; these relationships did not affect the work presented here. All other authors declare they have no competing interests.

**Acknowledgements**

This work was supported by the Children's Investment Fund Foundation, with continued funding from Clean Air Fund [grant numbers *1908-03995 and 341*]; with further funding support provided by Signe Ostby and Scott Cook (of the Valhalla Charitable Foundation), as well as funding by the Mayor of London for 10 additional AQMesh units. The Breathe London pilot project was convened by C40 Cities and the Mayor of London. The authors would like to thank the many hosts of
Breathe London monitors including local councils, schools, and residents, as well as the scientific and project advisors for their contributions. The authors are especially grateful to the local councils of Camden, Southwark, and Islington for continued access to reference monitors for collocations that were critical to this study. Thanks also to Greg Slater for his input on data visualization.



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
