# Peer review of "Evaluating uncertainty in sensor networks for urban air pollution insights"

_Atmospheric Measurement Techniques, 2021_

## Author Comment (AC1)

Reviewer comments are shown in italics, with author responses in normal text.

**Reviewer 1**

**General comments:**

*Congratulations for this huge and impressive work. The paper presents a very interesting use of low-cost sensors and sensor network within the scope of air quality monitoring with some innovative points. However, I was somehow disappointed going through the paper and seeing no data concerning the network calibration method while it as been described in the Methods paragraph and some of the conclusion are based on these particular results. Moreover, the title focus on the uncertainty evaluation while the paper use only RMSE and nRMSE, which, even if they gave relevant information about the quality of the data, I would not consider as an uncertainty but rather an error. From my point of view, through the whole document the word "uncertainty" is used in place of RMSE, nRMSE or error measurement. The author could maybe simply explain their choice of using the RMSE as an uncertainty evaluation tool.*

> We thank the reviewer for their feedback on the manuscript. We have responded to each of the specific comments below, and all revisions to the manuscript are described. We believe the revisions have improved the quality of the manuscript.

> Regarding the network calibration method, we have edited the text to avoid the impression of making conclusions about the method (our specific changes are listed in response to the Line 285-288 comment below). For the reviewer's second comment regarding the use of the term "uncertainty" when speaking about sensor measurement error, we have accepted the recommendation of the Editor ('EC1' in the open discussion) to continue using the term "uncertainty", which we discuss further in response to the Line 25 comment below.

**Specific comments:**

*Line 25: "average uncertainty (root-mean-square error)" why are you not directly speaking about RMSE, or error instead of uncertainty?*

> We appreciate the reviewer's question to clarify our use of the term "uncertainty" in the manuscript in relation to statistical metrics for sensor measurement error. In our view the term uncertainty is appropriate here as a general term to describe the possible deviation of sensor measurements compared to the true value. We specify root-mean-square error to be specific about how we are quantifying uncertainty in this case. Past literature evaluating sensor performance (e.g., Castell et al. 2017 https://www.sciencedirect.com/science/article/pii/S0160412016309989) has used the term uncertainty in the same way.

*Line 149: "excluding statistical outliers", how was this exclusion performed?*

> We have modified the following sentence in the methods to clarify this exclusion method (lines 150-152):

"Calibration gain and offset parameters were obtained by performing a linear regression on the hourly averaged collocation timeseries after excluding the 1st and 99th percentile of hours during the collocation based on the ratio of reference/candidate values."

*Line 224: "redaction" do you mean reduction?*

In this sentence we are referring to the process of redacting data based on invalid sensor flags, which affected the overall amount of valid data produced by the network. Redaction is the correct word.

*Line 285-288: "An evaluation of the performance of the independent network calibration method is included in Popoola et al. (in preparation). In brief, the estimated uncertainty of sensor measurements scaled with network method is broadly similar to the uncertainty of reference collocation-calibrated sensors (~30% median nRMSE)." Having not yet access to the performance evaluation results of the network method, it is rather difficult to conclude anything at this point. I hope the discussion will note insist on the network method.*

We appreciate the reviewer's comment and recognize that without presenting more detailed data about the novel network calibration method, it is not appropriate to make conclusions about its performance. In response we made several changes to the manuscript:

- We have added a citation to a conference presentation which describes the novel methodology in more detail and presents a preliminary evaluation of its performance during the BL project (line 161):

    Popoola, O.A.M. et al.: A novel calibration method for hyperlocal measurements of air quality using a low-cost sensor network, Air Sensors International Conference (ASIC): Virtual Fall Series, October 2020, available at: https://www.youtube.com/watch?v=sPzwmLNiP1w&ab_channel=UCDavisAirQualityResearchCenter, 2020.

- We have added text to our methods to clarify that the paper does not seek to present or evaluate the novel method, but rather describes it in sufficient detail to convey how some of the data was calibrated during the project (lines 162-165):

    "This manuscript does not intend to evaluate the network calibration method compared to other approaches. However, we describe the method here because it was used to scale a subset of BL sensors which had no physical (reference or transfer) calibration available, and we include data from this subset of sensors to maximize the number of sensor locations in our analysis and comparisons to the reference network."

- We have also removed the quantitative estimate of network method performance and added citation to the conference presentation containing a preliminary evaluation of network calibration method performance (lines 294-296):

"Preliminary evaluations have shown that the estimated uncertainty of BL sensor measurements scaled with the network calibration method is broadly similar to the uncertainty of reference collocation-calibrated sensors (Popoola et al., in preparation; Popoola et al., 2020)."

- We have removed suggestion of network calibration method on lines 298-300:

  "Enhanced QA/QC such as   calibration  on a near-continuous basis or seasonal bias corrections such as shown in Fig. S10 (see Sect. 3.2.1) could minimize variations in measurement uncertainty due to sensor performance."

- We have also made changes in the conclusions to avoid statements that cannot be supported by the data presented in the paper or distracting from the key messages of the manuscript (see response to Line 455-458 comment below).

*Line 288-289: "The results in Figs. 2 and 3 suggest that regardless of the method used for calibration, measurement uncertainty of sensors calibrated during a discrete period will be largely driven by the variability in the sensor performance over time." I don't see any comparison of calibration methods in the results of Fig. 2 and 3 which focused on colocation with reference methods.*

We appreciate the reviewer drawing attention to this statement which may not be conveying the intended message. The intent of this statement was to point out that error caused by temporal (seasonal) variability in sensor response would affect any sensor that is calibrated from a discrete short-term period. We have rephrased the sentence to ensure that we are not claiming to compare calibration approaches (lines 296-298):

"The results in Figs. 2 and 3 demonstrate that the long-term measurement uncertainty of sensors calibrated during a brief, discrete period is influenced by the changes in sensor response during different seasons and environmental conditions."

*Line 291: " S9 (see Sect. 3.2.1)" I don't see Fig. S8 cited before Fig. S9*

We thank the reviewer for noticing this error and have changed the order of the supplemental figures so that they are cited in order in the text.

*Figure 4: Is it possible to increase the number of ticks on the time axes instead of 1 tick every 6 months? It would greatly ease the understanding and help to follow the explanation given in the following paragraphs.*

We have increased the number of time labels to every 2 months in Figure 4 to make it easier to interpret.

*Line 307: "monthly mean NO2 concentrations", is there any reason why you compare monthly values instead of weekly or daily for example, even for this long term (2 years) trend? It is known that increase the time average length tends to smooth the sensors variation, decreasing the measurement error.*

We have added a sentence to explain why we decided to compare monthly values in Figure 4 (Lines 315 to 318):

> "We compare monthly values here to assess the sensor network's ability to reproduce long-term patterns observed by the reference network, on timescales that would be sensitive to effects of seasonal variations in pollutant concentrations or sensor performance, as well as long-term ambient pollution changes resulting from major interventions."

We agree that the monthly average will smooth out hourly, daily, etc. variation. However, the metrics discussed with this figure are the bias and trends in monthly mean concentrations which are supported by the figure. In addition, for the reasons mentioned by the reviewer, our performance evaluation metrics in Figs. 2 and 3 are calculated from the hourly timeseries data so the short-term variation in sensor performance is captured elsewhere in our error estimates.

*Line 424-425: "Additionally, the BL network could provide such information in near-real-time, making it a viable tool for rapid dissemination of air quality alerts." I would be more cautious about this conclusion as, if the scope is to give information about air quality alerts, a difference of roughly -15µg.m3 on a range of roughly 50 to 100 µg.m3 represent an error of -30 to -15% on the measurement value, which seems too large to be trusted for public information. Added the fact that the values are systematically underestimated.*

> We agree that the potential measurement error may be too great for trusted public information about the precise levels of air pollutant concentrations. However, the temporal profile of the sensor network is robust and closely tracks the timing of reference network-detected peaks and troughs. Therefore, we still believe the data can be valuable. However, how this information is communicated to the public is an area that we have not tried to answer in this manuscript. Because of this, we have decided to remove the statement.

*Line 455-458: "A cost-effective calibration approach such as the remote network calibration method can also be valuable for tracking and improving sensor performance over time by providing periodic calibrations and assessments of network performance, although a single point calibration was used here." It is difficult to acknowledge this conclusion based only on trust at this stage, as the results of the network calibration performance evaluation as not been really discussed in this paper, only by 1 sentence in paragraph 3.1.2.*

> The co-authors understand that there is not sufficient data about the network calibration method's performance presented in this manuscript to warrant presenting the method as an example and agree the statement should be modified. Therefore, we have revised the sentence to the following (lines 468-470):

> > "A near-real-time calibration approach may also be valuable for tracking and improving sensor performance over time by providing continuous calibrations and assessments of network performance, although a single point calibration was used here (Dye et al., 2021)."

This revised sentence is not focused on any single method, but instead suggests that a method which can adjust calibrations over time offers distinct advantages. This is supported by our observations (Figs. 2 and 3 especially) of sensor response varying over time after an initial calibration. We add a citation to Dye et al. as an example demonstration of a near-real-time calibration approach.

**Reviewer 2**

**General comments:**

*The manuscript describes the results from a deployment of up to 100 low cost sensor units across Greater London, focusing on data from NO2 by electrochemical cells and comparing them to the regulatory urban network. The study aims to address a question of interest for the scientific community, e.g. to what extent are low cost sensor units suitable for air pollution monitoring in urban areas? What might be the strength and weakness of a low cost sensor network? The study presents an extensive and impressive amount of work, whose publication will be beneficial for the scientific community.*

We thank the reviewer for their comments on the manuscript. We have addressed the specific comments below and noted our corresponding revisions, which we believe have improved the quality of the manuscript.

**Specific comments:**

*[line 275] "Aside from the seasonal variation in sensor bias and error, the initial calibrations seem to hold over the duration of the 18-month collocations." This sentence to me sounds almost self-contradictory. There is no standard for the definition of a "successful calibration" for LCS, but using the proposed benchmark of nRMSE < 50% and $R^2$ > 0.7, unit 83 at SK6 (figure 3c) is not within this range.*

We appreciate the reviewer pointing out this statement which may be confusing. The intent of the statement was to point out that the sensor bias followed a seasonal pattern such that the parameters were relatively similar a year later; not to state that the initial calibration is holding throughout the time when the sensor is deployed. We have removed this sentence to avoid confusion, seeing as a similar point is emphasized on lines 283-284 that explains the seasonal pattern of bias.

*The threshold in nRMSE and $R^2$, to be consistent, should be referred to the same time period (e.g. over 7-14 days for both long term collocations and other calibration methods); e.g. in figure 3b RMSE should be estimated over 7-14 days base to be comparable with figure 2. This might show that the initial calibration is not holding throughout the 18 months.*

We have adjusted Figure 3b to show MBE and RMSE over 14-day periods rather than monthly, and adjusted the text with corresponding 14-day period statistics to make them more comparable to Figure 2.

The updated Fig. 3b is below:

[Figure]

(b)

Figure 3: Performance of two calibrated sensors during long-term reference collocations. Sensors were calibrated using linear regression against the reference instrument during a two-week collocation directly preceding the evaluation period (calibration period not shown)... (b) MBE (eq. 1) and RMSE (eq. 2) statistics of hourly BL sensor measurements compared to reference measurements during 14-day periods…

The updated statistics (lines 281-283):

"…MBE of sensors during 14-day periods (Fig. 3b) ranges from -3 µg m$^{-3}$ to +11 µg m$^{-3}$ for unit 17 (-8% to +34% of the 14-day mean concentration) and from -8 to +19 µg m$^{-3}$ for unit 83 (-20% to +91% of the 14-day mean concentration)."

*paragraph 2.3.2 "Ozone cross-interference correction": the drift due to O3 interference is puzzling. According to Hossain et al (2016), the O3 scrubber should last at least for 14 ppm\*day (figure 4 Hossain et al (2016)). O3 hourly annual average at North Kensington for 2020 (data from https://londonair.org.uk/) is 28 ppb, resulting in ~7.4 ppm\*day, so there should not be any breakthrough over 6 months (line 171). I wonder if the calibration protocol, or the two stage calibration protocol (AQMesh + CERC), is playing a role in this instead of the O3 interference. How this could be checked without the access to the raw voltages of the cells?*

We appreciate the reviewer's comment and agree that it is challenging to rule out other factors, however based on our observations and analyses we suspect that ozone is the main driver of the issue. We agree that it is possible that the manufacturer's (AQMesh) proprietary calibration may be contributing to the apparent ozone effect, but such an evaluation was outside of the project scope (and the raw voltages were not logged). In response to this comment, we have provided additional evidence in support of the ozone cross-interference hypothesis (new supplemental figures S3 and S4). We have also updated the methods paragraph to state that the cross-interference was our hypothesis, but we could not rule out other issues, given the combination of factory and field calibration methods (lines 171-182):

"A long-term upward drift in BL $NO_2$ sensor measurements was observed (Fig. S2), which we hypothesized to be caused by an ozone cross-interference... Figs. S3 and S4 show evidence supporting the ozone cross-interference hypothesis and an evaluation of the correction method for an individual sensor. Note that due to a complex set of factors including the combination of factory (AQMesh) and field calibration methods, we couldn't exclude other possible causes of observed irregularities in sensor measurements."

Fig. S3 shows the apparent association between local ozone concentrations and sensor bias during one of the long-term collocations at a reference site with ozone measurements. The figure demonstrates that the effect appears to be present even during the first several months of the collocation.

Fig S4 shows the impact of $O_3$ on sensor $NO_2$ readings at one of the long-term collocation sites where we have both reference $NO_2$ and $O_3$. During periods where the reference $NO_2$ is low and reference $O_3$ is relatively high, uncorrected $NO_2$ sensor measurements are biased high and rarely register below 15 µg m$^{-3}$ even when reference concentrations approach 0 µg m$^{-3}$ (resulting in a "hockey stick" like scatter plot, Fig. S4 panels a & b). This unusual pattern is virtually accounted for by applying an ozone cross-interference correction (Fig. S4, panels c and d).

Lastly, while the reviewer is right that the average annual $O_3$ mixing ratio ~ 30 ppb for a typical urban site in London, it is worth stressing that we do observe significant seasonal variability, with summer-time highs of ~ 80 ppb (~ 160 µg m$^{-3}$; see Fig. S3). A cross sensitivity of as low as 10-20% at such high levels becomes significant particularly under low NOx conditions.

[Figure]

Figure S3: Monthly scatter plot of sensor percent bias without an ozone cross-interference correction applied vs. reference ozone measurement for a 17-month collocation of a BL sensor (Unit 83) at the SK6 (Elephant and Castle) LAQN reference site. Percent bias calculated as (uncorrected sensor $NO_2$ measurement – reference measurement) / reference measurement.

[Figure]

Figure S4: Time series and scatter plots of a BL sensor (Unit 83) at the SK6 (Elephant and Castle) LAQN reference site. (a) hourly timeseries and (b) scatter plot of BL sensor $NO_2$ without ozone cross-interference correction compared to reference $NO_2$. (c) hourly timeseries and (d) scatter plot of BL sensor $NO_2$ with ozone cross-interference correction compared to reference $NO_2$. BL sensor data in panels (a) - (d) was calibrated using linear regression against the reference instrument during a two-week period in May 2019, with (c & d) and without (a & b) an ozone cross-interference correction. (e) Reference $O_3$ at collocation site (SK6). Points in scatter plots (b) and (d) are colored by reference $O_3$ concentration, and a regression line and equation are shown to emphasize the large positive intercept in the absence of the correction. Ozone-corrected sensor data in panel (c) is a subset of data that was presented in Figure 3 of the manuscript.

*[lines 324-326] to me figure S8 points to a question: what is the lowest concentration which can be reliably measured with this network? Could it be reliably measured a 10 – 20 µg/m³ annual average of NO2?*

The reviewer's question is an interesting one. We can examine the error of sensors at collocations (same collocations as Fig. 2) with low (<20 µg m⁻³) mean $NO_2$ concentrations to quantify the average error during these less-polluted collocations. The following table presents the RMSE and nRMSE of collocations divided into two categories: those with mean reference concentrations above and below 20 µg m⁻³. The methodology is the same as that for Fig. 2, where the sensor evaluated has been scaled using linear regression results from a previous short-term collocation, with outliers occurring in July 2019 excluded.

| Collocation period mean NO$_2$ (μg m$^{-3}$) | Number of collocations | Median RMSE (μg m$^{-3}$) | Median nRMSE |
|---|---|---|---|
| > 20 μg m$^{-3}$ | 12 | 6.7 | 0.23 |
| < 20 μg m$^{-3}$ | 12 | 5.6 | 0.35 |

The RMSE of <20 μg m$^{-3}$ mean concentration collocations is 5.6 μg m$^{-3}$, compared to 6.7 μg m$^{-3}$ for all others. While this is a slightly lower RMSE in absolute terms, it is a larger proportion of the measured concentration, as indicated by the higher median nRMSE (0.35 vs. 0.23). This is an important point because when concentrations are very low or increments between sites are lower, the same sensor RMSE may not be tolerable compared to a more polluted location. We have added a point to the discussion (lines 485-487):

> "Furthermore, in another environment with different air pollution levels, the same magnitude of sensor RMSE may represent a different proportion of the average concentration, reinforcing the need to evaluate sensor performance locally and consider the tolerable amount of measurement error for each application."

*a minor point: in figures 4, 5, 6 it would help to have a bold dash horizontal line at 0 error for the lower panel, similarly to figure 3b*

We appreciate this suggestion and have added a dashed horizontal 0 line to the lower panels of Figs. 4-6 to help interpret the direction of the error in the plots.

*References*

*Marlene Hossain, John Saffell, and Ronan Baron, Differentiating NO2 and O3 at Low Cost Air Quality Amperometric Gas Sensors. ACS Sensors 2016 1 (11), 1291-1294 DOI: 10.1021/acssensors.6b00603*

**Additional Changes**

Below we note any additional revisions that have been made to improve the quality of the manuscript.

- Figure 1: The style of the scale bar was changed, a box was added around the inset of the UK map, and labels were added under Greater London and the United Kingdom respectively to improve the clarity and readability of the map
- Line 375: An erroneous reference to Fig. S6 was corrected to refer to Fig. 6

---

## Author Response (AR2)

**Response to Associate Editor**

**Peters et al., Evaluating uncertainty in sensor networks for urban air pollution insights (AMT-2021-210)**

In addition to our revisions detailed in the previous Response to Reviewers document, we have made the following changes in response to the Associate Editor's comments to the author:

*1) Please write in Section 2.3.1 explicitly that the network calibration method and its performance will be addressed in detail in a separate study (Popoola et al., in preparation).*

> We appreciate this suggestion and have added such a statement (Lines 163-164):

>> "The method and its performance will be addressed in more detail in a separate study (Popoola et al., in preparation)."

*2) In Section 2.6 it should be explained more clearly in the first sentence what exactly the ADMS model was used for. This becomes only clear towards the end of the section with the sentence "To calculate the modelled difference between BL and reference network means for the year 2019 ..".*
*ADMS was obviously used as a kind of transfer-standard between the two networks.*

> We thank the Editor for this clarification and have added a clause to the first sentence of Section 2.6 explaining the purpose of using the ADMS model (Lines 210-212):

>> "The ADMS-Urban air pollution dispersion model was used to simulate 2019 hourly $NO_2$ concentrations at BL and reference network monitoring locations in order to estimate the expected difference in $NO_2$ pollution levels between the two networks (McHugh et al., 1997)."